# Evaluation of School Children Nutritional Status in Ecuador Using Nutrimetry: A Proposal of an Education Protocol to Address the Determinants of Malnutrition

**DOI:** 10.3390/nu14183686

**Published:** 2022-09-06

**Authors:** Estephany Tapia-Veloz, Mónica Gozalbo, Gabriela Tapia-Veloz, Tannia Valeria Carpio-Arias, María Trelis, Marisa Guillén

**Affiliations:** 1Area of Parasitology, Department of Pharmacy and Pharmaceutical Technology and Parasitology, University of Valencia, 46010 Valencia, Spain; 2Department of Medicine and Public Health, Science of the Food, Toxicology and Legal Medicine, University of Valencia, 46010 Valencia, Spain; 3Research Group on Food and Human Nutrition, Escuela Superior Politécnica de Chimborazo, Riobamba 060101, Ecuador; 4Joint Research Unit on Endocrinology, Nutrition and Clinical Dietetics, University of Valencia-Health Research Institute La Fe, 46026 Valencia, Spain

**Keywords:** malnutrition, nutrition intervention, Nutrimetry, schoolchildren, intestinal parasites

## Abstract

The education sector is a cornerstone in the battle against malnutrition in children. However, there are still no consolidated protocols that outline strategies for how nutrition programs in low- and middle-income countries can be delivered through the education sector. Establishing the correct community diagnosis is essential prior to the elaboration of an intervention plan for a school population that takes into account more than just traditional variables related to the nutritional status. A total of 574 boys and girls aged 3–11 years from three educational institutions in different municipalities in Ecuador participated in the study. Sociodemographic, anthropometric (weight and height) and coproparasitological data were obtained. Nutrimetry, which is a combination of two classical anthropometrics indicators, was used for the analysis of the nutritional status, and the indicators’ frequencies varied among the schools. In order to improve the nutritional status of children, we proposed a framework mainly focusing on establishing alliances with the education sector and taking into account gender equality; respect for the environment; and the customs, beliefs and traditions of each population. The results obtained from the analyses of other variables demonstrated the importance of an adequate diagnosis prior to any type of intervention at the nutritional level, since characteristics could vary by local area and have an impact on the successfulness of the intervention.

## 1. Introduction

Malnutrition in childhood has an immediate impact on a child’s survival and development, and later on, it has an impact on their productivity and economic contribution to society [1,2]. The current agendas (the 2030 agenda, the global nutrition agenda, etc.) point out the importance of multisectoral approaches to addressing malnutrition [3,4,5]. The education sector is a cornerstone in the battle against malnutrition [6]. For this reason, programs that address the double burden of malnutrition (i.e., the existence of both undernutrition and overweight/obesity within the same population) [7] have been launched in low- and middle-income countries (LMICs) within the education sector (mainly in primary schools) [8,9]. The WHO’s Nutrition-Friendly Schools Initiative (NFSI) is one such initiatives that provides a framework for ensuring integrated school-based programs that address this double burden, taking into account the five broad components: school nutrition policies, awareness and capacity building of the school community, nutrition- and health-promoting curricula, supportive school environment for good nutrition, and supportive school nutritional and health services [8,9,10].

Despite these efforts, there are no consolidated protocols that outline strategies for how nutrition programs in LMICs can be delivered through the education sector. 

In Latin America, the coexistence of chronic undernutrition (stunting), and overweight and obesity in children is increasingly evident. It is estimated that the prevalence of chronic undernutrition in this region at the school stage is 34.4% [11], while the prevalence of overweight is 25% [12]. According to the 2018 Health and Nutrition Survey (ENSANUT) of Ecuador, the chronic undernutrition rate in this country for children under five years of age decreased from 23.9% to 23.0% between 2014 and 2018; in rural areas, this indicator decreased from 31.9% to 28.7% in the same period. The rate of overweight and obesity in children aged 5 to 11 years increased from 29.9% to 35.4%. Differences by area of residence were also observed; values were 36.9% in urban areas and 32.6% in rural areas [13]. These data show that, although the prevalence of chronic undernutrition is slowly decreasing, the prevalence of overweight/obesity is alarmingly increasing among children in Ecuador [14]. 

It can be observed that the so-called double burden of malnutrition in the case of this country is not alien to its reality, and it is increasingly worrisome [15]. It is directly related to the various changes in economic, social and demographic factors that occur where inequity and poverty are present, such as a higher migration from rural to urban areas; globalization; changes in occupational structures; alterations in dietary patterns; and a lifestyle characterized by an increased production, distribution and consumption of high-energy processed foods, along with a decrease in physical activity. All these elements together are causing a nutritional transition in countries such as Ecuador [14,15,16].

However, intestinal parasitism constituted by species *Giardia intestinalis*, *Ascaris lumbricoides* and *Trichuris trichuria* presents scientific evidence that proves it to be a determinant factor in the nutritional status of children [17,18,19]. Its importance lies in its high prevalence and its effects on the health status of the child population, such as a loss of appetite, growth retardation, cognitive problems, the poor absorption of nutrients and poor immunity. It has a negative influence on the morbidity and mortality of this population and thus on the economic development of a country [11,20,21,22]. The least-favored populations of structurally impoverished countries, such as Ecuador, have the highest prevalence due to multiple factors, such as poor environmental sanitation conditions, socio-cultural conditions, inadequate hygienic–sanitary habits and a high rate of maternal and infant undernutrition [11,20,21,22,23]. It affects approximately 80% of the rural population and 40% of the urban marginal population [24]. Studies carried out in this country show the prevalence of *G. intestinalis* to be between 16.4% and 20.3% [25,26], the prevalence of *A. lumbricoides* to be between 9.8% and 43.8% [25,27] and the prevalence of *T. trichuria* to be between 10% and 46.4% [28,29,30,31]. It is in second place on the list of the main causes of ambulatory morbidity devised by the Ministry of Public Health of Ecuador in 2014, and it is within the top ten causes of pediatric consultation [24,25,32]. 

In this scenario, it seems absolutely compulsory to carry out an evaluation and assessment of the childhood nutritional status, taking into account other variables rather than the classical ones, such as Body Mass Index-for-Age Z-Score (BMIZ) and Height-for-Age Z-Score (HAZ), before implementing a nutrition protocol.

For the diagnosis of children’s nutritional status at the population level, the BMIZ is an easy-to-use and -interpret indicator, but considering the nutritional transition that Ecuador is going through, it falls short. In view of this problem, the importance of a joint interpretation of the BMIZ with other indicators has been pointed out. This new approach has been implemented in individual care but not in epidemiological studies, which must be carried out before implementing a community intervention. For this reason, Solis and collaborators developed a method called “Nutrimetry”, which combines two simple and accessible anthropometric variables, BMIZ and HAZ, with the intention of facilitating their joint interpretation and generating a broader view of the nutritional status [33,34]. 

We undertook this research study to evaluate the nutritional status of school children from three different areas in Ecuador as the first step in order to develop proper nutritional protocols within the education sector. In this paper, we first describe the nutritional status in three areas using a new epidemiological methodology, Nutrimetry, based on the combination of two classical variables. We also propose a theoretical framework to develop in the education sector taking in account all the specific characteristics of the studied area in Ecuador. Moreover, we suggest some key strategies for how nutrition programs can be properly delivered through the education sector and how they can be strengthened in Ecuador. This framework attempts a link to the “One Health” breakthrough approach. According to the WHO, “One Health” is an approach to designing and implementing programs, policies, legislation and research in which multiple sectors communicate and work together to achieve better public health outcomes. It can be applied at the community, subnational, national, regional and global levels and relies on shared and effective governance, communication, collaboration and coordination [35,36]. We also discuss gaps in research, as well as opportunities for intervention in other places in Ecuador.

## 2. Materials and Methods

### 2.1. Study Design and Description of the Studied Local Areas

This cross-sectional study was conducted in three Ecuadorian primary schools located in three different provinces: two of them were located in Chimborazo, which belongs to the Sierra region, municipality of Penipe and Pallatanga, and the other was located in Guayas, which belongs to the Coast region, municipality of General Antonio Elizalde (Figure 1). Penipe has a cold mountainous climate; Pallatanga has a temperate sub-humid mountainous climate; and General Antonio Elizalde has a humid tropical climate.

According to the 2014 Map of Poverty and Inequality by Consumption in Ecuador, Pallatanga has a Gini coefficient of 0.37; Penipe has a Gini coefficient of 0.35; and General Antonio Elizalde has a Gini coefficient of 0.30 [37]. The years of schooling of the population in the Pallatanga municipality is 6.2 years; in Penipe, it is 7.7 years; and in General Antonio Elizalde, it is 8.3 years [38,39,40]. The percentage of households with inadequate housing characteristics in Pallatanga is 14.9%; in Penipe, it is 11.4%; and in General Antonio Elizalde, it is 5.7%. The percentage of homes with a public water supply in Pallatanga is 44.8%; in Penipe, it is 76.6%; and in General Antonio Elizalde, it is 63.2% [41]. These three municipalities differ from each other due to their location, climate, and socio-demographic and socioeconomic characteristics, thus allowing diverse results to be obtained for each area.

### 2.2. Recruitment and Sample

A primary school for each of the three municipalities was selected according to the following criteria: (1) number of students attending the schools had to be more than 300 and (2) with a lower to medium socioeconomic status (SES) to be representative of the population of the area. Finally, (3) the school staff had to be committed to and engaged in this preliminary phase of the project. To achieve this last point, informative talks addressed to the staff were given in each educational center. The pupils in each grade were recruited through their leaders (main teacher assigned for the school year), who informed them about the project during their lessons and gave some examples about how important nutrition is for mental and physical development. This process took two months. Meanwhile, all the parents were also invited to participate by the schools’ administrations, from whom they received an informative letter enclosed in the school diaries of their children. Informed consent forms signed by the parents of the participating students were obtained prior to data collection. No remuneration or incentive was provided for participation, but once the samples and anthropometric data were analyzed, a report with the results was delivered in a sealed envelope to their legal representatives, so that they could decide whether to proceed with medical assistance.

### 2.3. Ethics Approval

This study was approved by the Ethics Committee for Human Research of University of Valencia (procedure number: H1518738039128; 1 March 2018), by the Ethical Subcommittee for Research on Human Subjects of Central University of Ecuador (SEISH-UCE) (2019) and by National Directorate of Prevention and Control Strategies and National Directorate of Health Promotion of the Ministry of Public Health (MSP) (Ecuador, 2020) as part of the project Análisis de factores de riesgo y manejo de la malnutrición en edad escolar. Protocolo de actuación “aquí” para extrapolar “allí”. We ensured that the fundamental principles established by the Declaration of Helsinki and Spanish/Ecuadorian legislation in the field of biomedical research, data protection and bioethics were respected.

### 2.4. Data Collection

A group of ten nutritionists were trained to obtain data. We used standardized protocols to gather information concerning (a) anthropometric variables (weight and height); (b) sociodemographic variables (age, sex, educational level of the father and mother, whether the father/mother worked, where the bathroom was and bottled water consumption); and (c) coproparasitological variables (1 stool sample).
(a)Anthropometric Assessment

An Omrom^®^ electronic scale (accuracy, 100 g) and a Seca216^®^ mechanical measuring rod (accuracy, 1 mm) were used to determine weight and height. Height was determined without shoes and socks, and body weight was determined with participants dressed only in their underwear. HAZ and BMIZ were calculated using World Health Organization (WHO) Anthro and AnthroPlus software (World Health Organization, 2009; Anthro for Personal Computers, Version 3.01: Software for Assessing Growth and Development of the World’s Children), using the WHO child growth standard 2005 version for children aged 0–5 years and using the 2007 version for children and adolescents aged 5–19 years. These indicators classified children to varying degrees and types of malnutrition based on WHO reference data [42], which enable the analysis and interpretation of the growth patterns of children in any population in the world to be performed. For the final diagnosis of the nutritional status, Nutrimetry was used, where HAZ and BMIZ are combined in a 3 × 3 table (Figure 2) with the intention of facilitating their joint interpretation. 

The different combinations of HAZ and BMIZ result in nine categories, nutricodes, which are assigned nine-digit codes, 1, 3, 4, 5, 6, 7, 8, 9 and 11 (Table 1); these are responsible for representing the different diagnoses of the nutritional status [33].
(b)Sociodemographic variables

A questionnaire was used to collect personal data concerning the following:

(1) Date of birth, (2) sex, (3) education level of the father, (4) education level of the mother, (5) whether the father/mother worked, (6) where the bathroom was and (7) bottled water consumption. Items three to seven were used to assess the socioeconomic status level of the family.
(c)Coproparasitological Assessment

A total of 370 samples were collected and processed. One fresh stool sample was collected per participant. After the filtration and concentration of the samples, they were analyzed using the Kato–Katz technique and optical microscopy for the identification of forms of resistance of intestinal parasites in general (cysts and eggs); molecular diagnostic techniques (real-time polymerase chain reaction or qPCR) were also applied to identify one specific parasite (*Giardia intestinalis*). 

Starting with 3 g per fecal sample, they were filtered and concentrated via centrifugation (2500 rpm, 5 min) in Midi Parasep tubes^®^ (Apacor Ltd., Wokingham, UK). The sediment obtained was divided into two microtubes, one for the extraction of total stool DNA for qPCR and the other with 10% formalin for microscopic observation. QIAmp DNA Stool Mini Kit (QIAGEN^®^, Hilden, Germany) was used for DNA extraction according to the manufacturer’s instructions.

For the diagnosis of *G. intestinalis* DNA, qPCR was performed. This protocol specifically amplifies a fragment of the gene that encodes the parasite’s small ribosomal subunit RNA (SSU rRNA) and shows high sensitivity [43]. A commercial assay with the specific primers and probe (LightMix Modular Assays Giardia; Roche^®^, Basel, Switzerland) was used, together with mastermix (dNTPS; thermostable Taq polymerase and buffer) (PerfeCTa qPCR ToughMix; Quanta Biosciences, Gaithersburg, MD, USA) for a mix reaction final volume of 15 μL. To each well, 5 μL of sample and positive control DNA was added, and water was added instead for the negative control. The analysis was performed with StepOnePlus real-time PCR thermocycler^®^ (Applied Biosystems^®^, Foster City, CA, USA). Any sample that managed to amplify before 43 cycles was considered positive.

### 2.5. Data Analysis

Descriptive statistics were calculated, including measures of central tendency (mean and median), measures of dispersion (standard deviation, range and coefficient of variation) and measures of shape (asymmetry and pointing) for quantitative variables, as well as the absolute and relative frequencies for the qualitative variables. Association analyses were performed, stratifying by sex and age to observe possible heterogeneity in the results according to these factors. When data were stratified and the sample size was small (n < 15), non-parametric tests were used (Fisher’s exact test and Mann–Whitney U test). Any *p*-value less than 0.05 was considered statistically significant. All the variables were analyzed using SPSS software (Statistical Package for Social Sciences for Windows, version 26.0; SPSS Inc., Chicago, IL, USA).

### 2.6. Theoretical Framework for the Implementation of Nutritional and Health Education in Ecuador

Figure 3 shows an overview of the main components of a comprehensive framework for the implementation of an adequate nutrition program in Ecuador taking in account the five broad components of the NFSI. This framework is based on there being accessible primary schools in the areas where an intervention must be carried out. Schools are the cornerstone, as they serve as the vehicle by which nutrition programs are delivered. 

The development process consists of the following four phases:
Phase 1—Diagnosis: In this phase, Nutrimetry plays an important role, but it is not sufficient to improve the nutrition of vulnerable populations in food-insecure areas. For this reason, we propose six components on which information must be collected in order to establish a diagnosis. These six components are as follows: (1) school services, (2) socioeconomic status [44], (3) Nutrimetry, (4) agriculture resources [45], (5) water sanitation [11,20,21,22,23] and (6) myths, customs and traditions related to nutrition and/or health.Phase 2—Identification of health-promoting resources at schools: Before implementing an intervention, it is important to identify any health-promoting resources. These resources promote health and facilitate coping with stressors, such as social relationships and dietary practices. To facilitate the mobilization of health resources, intervention strategies should be adjusted to real life in order to increase the chance of the successful implementation of newly adopted behaviors in everyday life [46].Phase 3—Intervention: In Appendix A, an approach for an intervention to improve the nutritional status in Ecuador is shown. The six components mentioned in Phase 1 (Diagnoses) and some of the variables presented in Phase 2 (Identification of health-promoting resources at schools) are taken into account in order to establish intervention objectives, indicators, verification sources and activities.
Phase 4—Evaluation outcomes: Nutrition indicators should be included in any intervention to assess the improvement of the nutritional and health status. However, we also must include other indicators related to the consumption of local and nutritious food.

## 3. Results

A total of 574 schoolchildren participated, of whom 304 were girls (53%) and 270 were boys (47%). The mean age was 7.7 ± 2.3 years. We recruited 297, 143 and 134 schoolchildren in Pallatanga, General Antonio Elizalde and Penipe, respectively.

The number of the total students of the three educational centers, the ratio of participation and the frequencies (%) of the studied population according to gender and the age group (<5 years, 5–8 years and ≥9 years) stratified by the three municipalities (General Antonio Elizalde, Pallatanga and Penipe) are shown in Figure 4.

Table 2 shows the following sociodemographic variables stratified by municipality: father’s educational level, mother’s educational level, whether the father or mother worked, indoor bathroom and bottled water consumption. We found statistically significant differences in the frequencies of all variables among the three municipalities (*p* < 0.05). General Antonio Elizalde presented figures that could be associated with an economic level that is higher than those of Penipe and Pallatanga, and this is consistent with the Gini coefficient described by the National Institute of Statistics and Census (INEC) in its document Map of Poverty and Inequality by Consumption in Ecuador 2014 [37]. These differences could especially be seen in the educational levels of the father and mother, bathing inside or outside of the house and bottled water consumption; in General Antonio Elizalde, the university level of education of the father and mother exceeded 30%, while in Pallatanga and Penipe, it did not even reach 9%. In General Antonio Elizalde, bathing inside the house occurred in almost all households (92.3%), while in Pallatanga and Penipe, it only occurred in approximately half of the population. There was a similar trend regarding bottled water consumption; in General Antonio Elizalde, 67.8% of participants said they consumed bottled water, while in Pallatanga and Penipe, this value was much lower, at 16.2% and 16.4%, respectively. 

The results of the coproparasitological analysis are shown in Table 3. Overall, 38.6% of the participants presented intestinal parasites. *G. intestinalis* was the most prevalent parasite (30.8%), which was analyzed using light microscopy and qPCR to improve its sensitivity. *A. lumbricoides* was the second most prevalent parasite (9.5%), and *T. trichiura* was the third most prevalent parasite (5.2%); for both helminths, no molecular method was chosen because the Kato–Katz technique was optimal. Most of these parasitic species are ingested as cysts or eggs (infectious forms), and they are present in food or water as a result of fecal contamination. When stratified by municipality, being parasitized or having *G. intestinalis* did not present a statistically significant difference (*p* > 0.05), but it did in the cases of *A. lumbricoides* and *T. trichiura*. Pallatanga had the highest figures of 16.1% and 9.3%, respectively, while the other municipalities presented lower data for the two helminths, which did not exceed 4.4% for *A. lumbricoides* or 2.1% for *T. trichiura*. The reasons for these differences are unknown and should be investigated in future research.

Regarding nutritional status, which was evaluated using Nutrimetry, the total data stratified by gender and age are shown in Table 4. The prevalence values of the nutricodes associated with malnutrition (nutricodes 9, 4 and 3) were 27.2%, 10.6% and 4.9%, respectively. Nutricode 5 was non-existent. We did not find any statistical difference (*p* < 0.05) between boys and girls stratified by age in the distribution of the nutricodes. 

We analyzed the distribution of the nutricodes of all of the schoolchildren in each municipality, so there were differences in some variables associated with socioeconomical status that could determine differences in nutritional status. The data are shown in Table 5.

## 4. Discussion

This paper attempts to establish the basis for a protocol to address the determinants of malnutrition in Ecuador using schools and education as the cornerstones. To our knowledge, this is the first time this approach has been used in the country. The data collected are part of a project that has just been awarded in the call for International Cooperation Projects 2022 of University of Valencia, where the intervention phase is planned to begin in January 2023; all the data presented in this article allowed us to have an overview of the reality and lay the groundwork for developing the proposed framework, which we are using with the matrix to prepare all the tools and materials to be used in the intervention.

We present the theoretical framework for the implementation of nutritional and health education in Ecuador and also the results of the analysis of some of the components of the theoretical framework. Although the authors are aware that only a small part of the components of the theoretical framework were analyzed, these preliminary results are really interesting and need to be analyzed before moving on to the implementation of the protocol, which is to be specific to each community.

We used Nutrimetry to assess the nutritional status of children in three areas of Ecuador. Nutrimetry, as well as the assessment of some other determinants, provides relevant epidemiological information to establish protocols for future nutritional interventions in schoolchildren. The prevalence of overweight (codes 7, 9 and 11) was 29.2%; the prevalence of thinness (codes 1, 3 and 5) was 6.3%; and the prevalence of short stature (codes 1, 4 and 7) was 14.3%. Similar figures were found in other studies in this Ecuadorian school population [25,47]. Our study analyzed three schools in three different municipalities, which differ due to their location, altitude, climate, population and economic characteristics. The latest published official data on the Gini coefficient for the municipalities are from 2014; there are more current studies but not at the municipal level [48,49]. According to this coefficient, which allows wage inequality to be measured, Pallatanga had the highest inequality (0.37), followed closely by Penipe (0.35) and finally by General Antonio Elizalde (0.30) [37]. When we determined whether there was an association between the area in which the schoolchildren lived and nutricodes, a statistically significant difference was observed (*p* < 0.001). Accordingly, Roche et al. (2016) stated that a high socioeconomic level decreases the risk of stunting [50] but also increases the rates of overweight [51]. This was the case in General Antonio Elizalde, where overweight and obesity (codes 7, 9 and 11) reached 42.7% and were much more prevalent than some types of undernutrition; even code 1 (low height and thinness) was nonexistent. Pallatanga and Penipe, which have a lower socioeconomic level than General Antonio Elizalde, had lower figures for excess weight, but they were still high, at 31.3% and 11.9%, respectively. As for thinness (codes 1, 3 and 5), Pallatanga had a rate of 4.7%, and Penipe had a rate of 11.2%. According to these results, Penipe was the municipality with the lowest figures for excess weight and the highest figures for thinness; however, according to the Gini coefficient [37], its economic level is slightly higher than that of Pallatanga. This could be due to the fact that its socioeconomic growth may have changed in recent years, and Penipe is still a cold mountainous area at a high altitude with a large indigenous population; it has always had the highest percentages of malnutrition [50,52]. All the results presented show that a structurally impoverished country such as Ecuador, which is normally associated with high levels of child undernutrition, also currently presents alarming rates of excess weight. It is true that the trend of child undernutrition has been reduced, but this is still insufficient; thus, it is a latent problem that has not been eradicated. Therefore, there is an evident nutritional and epidemiological transition [14,53,54,55]. The problem is that any type of malnutrition during childhood is associated with adverse health consequences throughout life [56], affecting morbidity, mortality, the development of skills, educational performance, social inclusion, labor, the environment and productivity [57]. The reasons for the presence of an altered nutritional status are intimately linked to complex socioeconomic factors, such as changes in occupational structures; rapid urbanization; an increased penetration of the retail food industry, which has resulted in diets based on energy-dense and nutrient-poor foods; maternal education; and economic growth, which is often accompanied by persistent poverty and inequality, a reality not alien to that of Ecuador, where there is high economic inequity that is negatively affecting the nutritional status of its population [14,51,58,59]. To the best of our knowledge, this is the first time that Nutrimetry was used to assess the nutritional status in Ecuador. Previous studies only used the BMIZ indicator, the HAZ indicator or both indicators but analyzed them separately [13,14,50,51,52,53,59], so children suffering from malnutrition may not have been detected. In the case of the present study, 82 children with stunting (codes 1, 4 and 7) would have been left out, as well as 28 children who presented a recent alteration in nutritional status (code 3). Nutrimetry allows us to describe and analyze the population distribution of malnutrition in greater detail and depth. Nine subgroups are generated without the requirement of any specialized software for processing. It also complies with WHO criteria in order to be considered an anthropometric indicator, using its growth standards for interpretation. Furthermore, it discourages the stigmatization of individuals using neutral language since it reports results in numbers, avoiding the use of words such as “fat” and “skinny”, thus facilitating the communication of the results and objectives to schoolchildren without an emotional semantic load [33,34].

Intestinal parasitism is an important factor for the correct epidemiological analysis in the case of countries such as Ecuador, as such countries are characterized by high numbers of intestinal parasitism, especially in children [60,61]. There is evidence that certain types of intestinal parasites, such as *G. intestinalis*, *A. lumbricoides* and *T. trichiura*, have an impact on nutritional status [17,18,19], and in the case of Ecuador, previous research has described significant prevalence of these parasites [25,26,27,28,29,30]. After an analysis of the samples collected, we found that 38.6% of the children were parasitized with any one of these three species. *G. intestinalis* was the most prevalent at 30.9%; this figure is higher than that reported in previous research [61,62,63], because we used, in addition to microscopy, real-time PCR, which is a more sensitive technique [64]. *A. lumbricoides* and *T. trichiura* had a prevalence of 9.5% and 5.1%, respectively, and other articles have reported similar figures [25,60,64]; to detect these helminths, we did not use any molecular technique, since optical microscopy together with the Kato–Katz technique is optimal for their detection. When analyzing the relationship between Nutrimetry and being parasitized, we did not find a statistically significant association, possibly because the nutritional impact is usually related to undernutrition, and in our study, such prevalence was not high. However, it is still a problem, since the number of intestinal *G. intestinalis* cases was high; it is possible that many of these children were asymptomatic [65,66] and that it is a triggering factor for secondary food intolerance/malabsorption, dyspepsia or irritable bowel syndrome [19,67,68]. It is known that it causes the activation of CD8 lymphocytes in the intestinal villi of the absorptive mucosa, affecting the growth and cognitive development of children suffering from it [65,69,70]. Studies have also associated *G. intestinalis* with alterations in iron absorption, decreased serum iron, low blood Hb levels and iron-deficiency anemia [19,65]. As for helminths, these cause reduced growth rates, intellectual disabilities, cognitive deficits, decreased food intake, iron deficiency and the malabsorption of nutrients [17,27]. The relationship between the studied municipalities and being parasitized was not significant in the case of *G. intestinalis* (*p* > 0.05), but it was in the cases of *A. lumbricoides* and *T. trichiura* (*p* < 0.05), where Pallatanga had a higher prevalence than did Penipe and General Antonio Elizalde. These disparities may be due to the fact that the numbers of these two parasites vary greatly depending on the area studied; previous studies have reported values ranging from 10% to 46.8% [25,27,28,29,30]. It is necessary to further investigate the causes of these differences. The figures for the parasitic infections of *A. lumbricoides* and *T. trichiura* were low with respect to *G. intestinalis*, and this was probably due to the deworming campaigns carried out in schools for helminths but not for protozoa. The prevalence of these three pathogens may be associated with insufficient hygienic–sanitary habits, inadequate environmental conditions, and insufficient resources and education, which favor their development, maintenance and dissemination [71].

Ecuador, unfortunately, despite the policies, strategies and programs implemented in recent years, such as Plan Nacional del Buen Vivir, Plan Intersectorial de Alimentación y Nutrición, Intervención Nutricional Territorial Integral (INTI) and Programa Acción Nutrición (PAN) [57,72,73] and the incorporation of healthy habits in the curriculum and the increase from two to five hours of physical education for public schools and colleges [73,74], has not achieved a substantial improvement in school malnutrition figures. This fact is evidence that there is much work ahead and that Ecuador should be focused on malnutrition in the forms of deficit and excess, taking into account the nutritional epidemiological transition that is taking place. For the elaboration of the framework and for the initial evaluation of the school nutrition situation in our study, the NFSI tools [9] were taken into account because of their great relevance and usefulness in this population. The results of this study make it clear that there is an unquestionable need for intervention programs; therefore, the framework we propose as well as the intervention planning matrix (Appendix A) could help in solving the current problems of schoolchildren malnutrition in Ecuador.

Finally, we would like to point out that given the volume and complexity of the information that researchers could obtain when this approach is implemented in other municipalities in Ecuador, it will likely be necessary to use informatics tools and integrated platforms. Nutritional Informatics (NI) could provide some valuable tools [75] and help to develop solutions to the health problems encountered within a transitioning food environment. The definition of NI expanded as an effective retrieval, organization, storage and optimum use of information, data and knowledge for food and nutrition-related problem solving; Informatics is supported using information standards, information processes and information technology. On the other hand, the importance of integrating social factors not only in nutritional and health education but also in medical care is hardly news [76]. Social and other “non-medical” determinants of health are also obtained and then digitally integrated. Acknowledging, measuring and addressing all these data into the mainstream community care context is essential to achieve true health and wellbeing among the populations living in our neighborhoods, regions and nations.

The present research study had some limitations; due to the non-representative study sample, the data obtained and analyzed from the three schools could not be extrapolated to the reality of the municipality or the country; a larger sample of participants is needed to be representative, but in any case, we showed that school malnutrition is a reality that should be studied and solved. The leaders notified all the students enrolled to participate, but the rate of commitment varied for each educational center, being the lowest in Penipe, possibly due to the characteristics of its population, since many of the children lived in remote rural areas and for their parents to travel was to lose a day of work, which is something important to take into account in order to increase the degree of commitment as well as to investigate the reasons for low attendance. We only analyzed the nutritional status, parasitic infection and some items related to socioeconomic level, which are important, but it is necessary for future research to take into account the other variables proposed in the framework, as well as the collection of other health parameters to detect micronutrient deficiencies in order to have the best possible diagnosis prior to intervention. In addition, for an in-depth study of the socioeconomic level in Ecuador, there are validated questionnaires from the INEC [77] that collect many more data and that should be considered. The approximate costs derived from the implementation of the phases of the framework were not calculated; therefore, we do not know the necessary budget, and for the following studies, it would be relevant to establish this factor prior to its application and to confirm its feasibility.

## 5. Conclusions

The first step that must be taken before developing a nutritional intervention plan for the pediatric population is establishing the correct diagnosis of the current status of the area related to all the determinants of malnutrition. Nutrimetry can be used as a suitable indicator at the epidemiological and clinical levels, as it allows data to be obtained by combining two variables that are normally used separately, such as HAZ and BMIZ, and it is easy to use and interpret through the nutricodes generated, as well as being useful in the evaluation phase. It is important that the intervention to be carried out identifies the health-promoting resources adjusted and adapted to the reality of the population by taking into account the NFSI tools. The establishment of alliances with local centers, institutions and associations, whether public or private, would allow sustainability to be maintained, continuity to be ensured and self-sufficiency to be generated.

## Figures and Tables

**Figure 1 nutrients-14-03686-f001:**
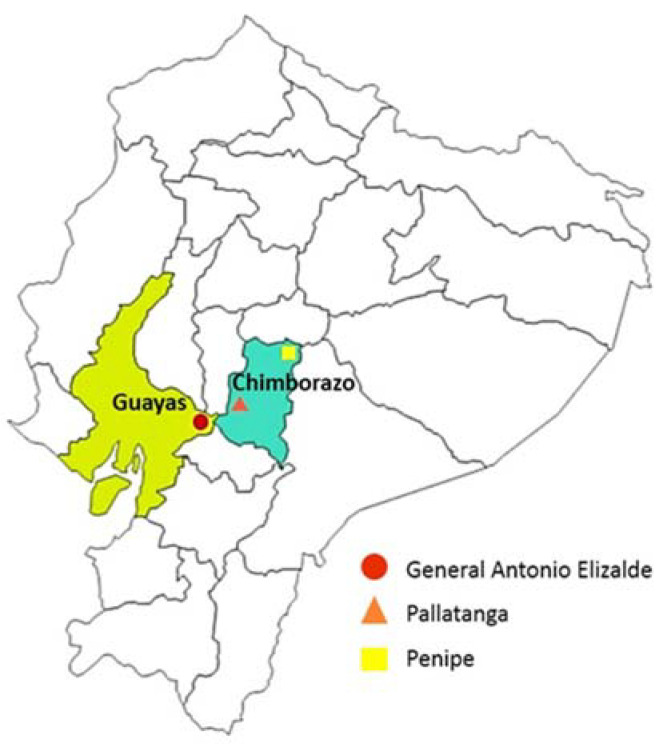
Map of Ecuador showing localization of the three municipalities in the Chimborazo and Guayas provinces.

**Figure 2 nutrients-14-03686-f002:**
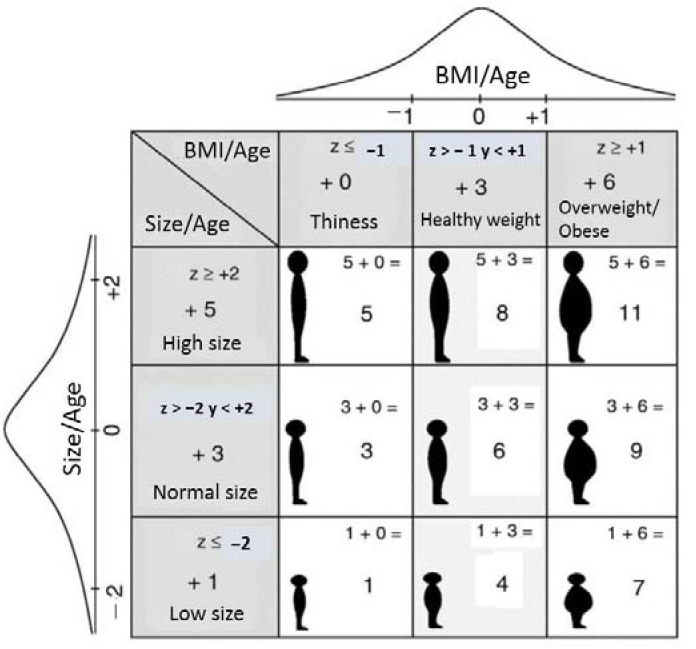
Nutrimetry (taken and translated from Selem-Solis et al. (2017).

**Figure 3 nutrients-14-03686-f003:**
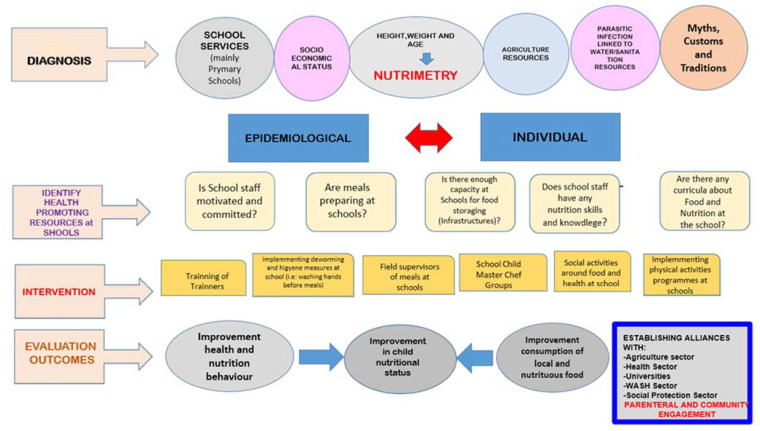
Theoretical framework for the implementation of nutritional and health education in Ecuador focused mainly on primary school environments.

**Figure 4 nutrients-14-03686-f004:**
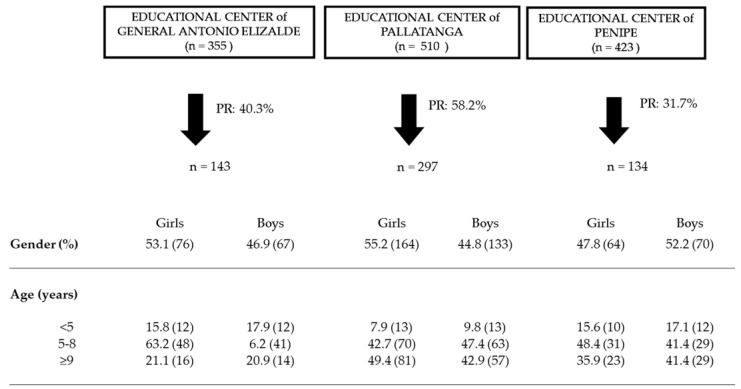
Participation rate (PR) and frequency (%) of the studied population according to gender and age group in the three municipalities. Data are shown as percentages of the total population stratified by municipality. The number of people is shown in brackets. There are no statistically significant differences (*p* > 0.05) for the distribution of the frequencies of participants by age and gender between the three municipalities.

**Table 1 nutrients-14-03686-t001:** Nutricodes and their interpretations.

Nutricodes	Interpretation
1	Low HAZ + Low BMIZ
3	Normal HAZ + Low BMIZ
5	High HAZ + Low BMIZ
4	Low HAZ + Normal BMIZ
6	Normal HAZ + Normal BMIZ
8	High HAZ + Normal BMIZ
7	Low HAZ + High BMIZ
9	Normal HAZ + High BMIZ
11	High HAZ + High BMIZ

HAZ, Height-for-Age Z-Score; BMIZ, Body Mass Index-for-Age Z-Score.

**Table 2 nutrients-14-03686-t002:** Frequencies (%) of the studied population according to education level of the father, education level of the mother, whether father or mother worked, indoor or outdoor bathroom and bottled water consumption stratified by municipality.

		General Antonio Elizalde	Pallatanga	Penipe	Total	* *p*-Value
		(n = 143)	(n = 297)	(n = 134)	(n = 574)	
Education level of the father	no studies	0 (0)	7.2 (17)	12.9 (16)	6.6 (33)	0.000
	elementary school	10.0 (14)	57.0 (135)	54.8 (68)	43.3 (217)	
	high school studies	55.7 (78)	31.2 (74)	24.2(30)	36.3 (182)	
	university studies	34.3 (48)	4.6 (11)	8.1 (10)	13.8 (69)	
Education level of the mother	no studies	0 (0)	6.3 (18)	12.7 (17)	6.2 (35)	0.000
	elementary school	8.4 (12)	51.0 (146)	50.7 (68)	40.2 (226)	
	high school studies	55.2 (79)	36.7 (105)	32.1(43)	40.3 (227)	
	university studies	36.4 (52)	5.9 (17)	4.5 (6)	13.3 (75)	
Father or mother worked	yes	94.2 (131)	89.7 (209)	96.8 (120)	92.7 (496)	0.036
Indoor or outdoor bathroom	indoor	92.3 (132)	53.9 (160)	43.3 (58)	61.0 (350)	0.000
Bottled water consumption	yes	67.8 (97)	16.2(48)	16.4 (22)	29.0(167)	0.000

Data are shown as percentages of the total population stratified by municipality. The number of people is shown in brackets. * *p*-value for the comparison of distribution of the frequencies among the three municipalities.

**Table 3 nutrients-14-03686-t003:** Frequencies (%) of the studied population according to the presence of total parasitic infection and infection by *G. intestinalis*, *A. lumbricoides* and *T. trichiura* stratified by municipality.

	General Antonio Elizalde	Pallatanga	Penipe	Total	* *p*-Value
	(n = 114)	(n = 161)	(n = 95)	(n = 370)	
Parasitic Infection	37.7 (43)	41.6 (67)	34.7 (33)	38.6 (143)	0.478
*G. intestinalis*	34.2 (39)	27.3 (44)	32.6 (31)	30.8 (114)	0.452
*A. lumbricoides*	4.4 (5)	16.1 (26)	4.2 (4)	9.5 (35)	0.006
*T. trichiura*	1.8 (2)	9.3 (15)	2.1 (2)	5.2 (19)	0.001

Data are shown as percentages of the total analyzed samples of the population stratified by municipality. The number of people is shown in brackets. * *p*-value for the comparison of the distribution of the frequencies among three municipalities.

**Table 4 nutrients-14-03686-t004:** Frequencies (%) of the studied population according to codes established for Nutrimetry stratified by gender and age group (<5 years, 5–8 years and ≥9 years).

	BOYS (n = 270)			* *p*-Value	GIRLS (n = 304)			* *p*-Value	** *p*-Value
	<5	5–8	≥9		<5	5–8	≥9		
	(n = 37)	(n = 133)	n = (100)		(n = 35)	(n = 149)	n = (120)		
Thinness				0.140				0.482	0.71
Nutricode									
1	0 (0)	0.8 (1)	1.0 (1)		0 (0)	0.7 (1)	4.2 (5)		
3	2.7 (1)	8.3 (11)	4.0 (0)		11.4 (4)	2 (3)	4.2 (5)		
5	0 (0)	0 (0)	0 (0)		0 (0)	0 (0)	0 (0)		
Normal Weight									
Nutricode									
4	16.2 (6)	9.8 (13)	14 (14)		11.4 (4)	6.7 (10)	11.7 (14)		
6	62.2 (23)	55.6 (74)	50 (50)		54.3 (19)	53 (79)	49.2 (59)		
8	0 (0)	0 (0)	0 (0)		0 (0)	1.3 (2)	0.8 (1)		
Overweight and Obesity									
Nutricode									
7	2.7 (1)	1.5 (2)	6.0 (6)		2.9 (1)	0.7 (1)	1.7 (2)		
9	16.2 (6)	24.1 (32)	25.0 (25)		20 (7)	35.6 (53)	27.5 (33)		
11	0 (0)	0 (0)	0 (0)		0 (0)	0 (0)	0.8 (1)		

Data are shown as percentages of the total population stratified by age and gender. The number of people is shown in brackets. * *p*-value for the comparison of the distribution of the frequencies of the codes established by Nutrimetry and age ** *p*-value for the comparison of the distribution of the frequencies of the codes established by Nutrimetry and sex.

**Table 5 nutrients-14-03686-t005:** Frequencies (%) of the studied population according to codes established for Nutrimetry stratified by municipality.

	General Antonio Elizalde	Pallatanga	Penipe	Total	* *p*-Value
	(n = 143)	(n = 297)	(n = 134)	(n = 574)	
Thinness					0.000
Nutricode					
1	0 (0)	1.3 (4)	3.0 (4)	1.4 (8)	
3	4.9 (7)	3.4 (10)	8.2 (11)	4.9 (28)	
5	0 (0)	0 (0)	0 (0)	0 (0)	
Normal Weight					
Nutricode					
4	2.8 (4)	11.1 (33)	17.9 (24)	10.6 (61)	
6	49.7 (71)	52.2 (155)	58.2 (78)	52.9 (304)	
8	0 (0)	0.7 (2)	0.7 (1)	0.5 (3)	
Overweight and Obesity					
Nutricode					
7	0.7 (1)	3.4 (10)	1.5 (2)	2.3 (13)	
9	41.3 (59)	27.9 (83)	10.4 (14)	27.2 (156)	
11	0.7 (1)	0 (0)	0 (0)	0.2 (1)	

Data are shown as percentages of the total population stratified by municipality. The number of people is shown in brackets. * *p*-value for the comparison of the distribution of the frequencies of the codes established by Nutrimetry among the three municipalities.

## Data Availability

The data sets generated or analyzed during the current study are available from the corresponding author upon reasonable request.

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
