# Peer review of "Evaluation of School Children Nutritional Status in Ecuador Using Nutrimetry: A Proposal of an Education Protocol to Address the Determinants of Malnutrition"

_nutrients, 2022, doi:10.3390/nu14183686_

Round 1
Reviewer 1 Report (Previous Reviewer 1)
Dear authors, the edits to the paper do not sufficiently address the comments presented in the review. I kindly suggest the comments are re-considered.
Author Response
NUTRIENTS 1873439
Evaluation of School Children Nutritional Status in Ecuador using Nutrimetry. A Proposal of an Education Protocol to Address the Determinants of Malnutrition
Reviewer 1
Dear authors, the edits to the paper do not sufficiently address the comments presented in the review. I kindly suggest the comments are re-considered.
Dear Sir or Madam, first of all authors would like to thank you for your contribution to improve our manuscript.
The most relevant and important point that a previous reviewer has mentioned about our paper is that “the paper is incomplete” and “at least all components of Phase 1 of the process should be presented in the paper, which is not the case”. To address this important comment and clarify the objective of our manuscript we mention in the first paragraph of the Discussion Section:
“This paper attempts to establish the basis for a protocol to address the determinants of malnutrition in Ecuador using School and Education as the cornerstone. To our knowledge, this is the first time this approach has been used in the country. The data collected are part of a project that has just been awarded in the call for International Cooperation Projects 2022 of the University of Valencia, where the intervention phase is planned to begin in January 2023 and all the data presented in this article has allowed us to have an overview of the reality and lay the groundwork for developing the proposed framework, which we are using with the matrix to prepare all the tools and materials to be used in the intervention.
We present the theoretical framework for the implementation of nutrition and health education in Ecuador and also the results of the analysis of some of the components of the theoretical framework. The development process consists of five phases, but we only focus our work on some of the components of Phase 1 (Diagnosis), mainly the component called Nutrimetry. Socioeconomic status and parasitic infection linked to water sanitation resources are also partially explored. Although the authors are aware that only a small part of the components of the theoretical framework have been analyzed, these preliminary results are really interesting and need to be analyzed before moving on to the implementation of the protocol, which will be specific to each community.”
Authors consider that these two paragraphs explain the purpose of our preliminar work that is an important part of a wider project that will carried out in next years in Ecuador. We would like to point out that the original title of the manuscript was also modified to clarify the objective of the manuscript (before it was “Integrating nutrimetry as a prediagnosis evaluation in a holistic education protocol to address the determinants of malnutrition in Ecuador”). However, if the reviewer actually consider that this changes and /or modifications are not sufficiently address this part of the reviewer’s comments, please let us know what the main amendments need to be performed.
Another important comment addressed by the previous reviewer is related to concepts or strategies that are mentioned but not encompassed in the manuscript:
- ONE HEALTH
- THE NUTRITION FRIENDLY SCHOOL INITIATIVE (NFSI)
- GENDER, ENVIRONMENT AND CULTURE'
All these concepts and general strategies appear in the proposed theoretical framework (Figure 3) and also they are developed as specific “Activities” in the Table S1 (Intervention planning matrix with intervention objectives, indicators, sources of verification and activities focused primarily on the elementary school setting)
For example:
Activities related to ONE HEALTH
“Generate written agreements with the local authorities stating that they will collaborate in order to guarantee their participation” or
“Take measures to improve sanitation in the area by the municipality and ensure equitable access to drinking water”
Activities related to (NFSI)
“Include classes on healthy habits in at least one subject of the primary education curriculum” or
“Ensure that, in physical education, all students interact and that different activities and sports are included to ensure that they are motivated and that their needs adapted to”
Activities taking into account gender, environment and culture
“Provide informative talks and interactive workshops adapted to each population according to their level of knowledge” or
“Form school cooking groups, “Master chef”, where students are taught how to prepare healthy meals with seasonal and local foods”
Some of these specific “Activities” could cover more than one concept or general strategies
The authors would like to insist again on the fact that the manuscript contains a proposal to implement a nutritional program in schools in Ecuador taking into account the specific features of every educational center. This proposal will be carried out in the next few years. To date, part of one of the 5 phases of the proposal has been carried out, but it has already produced interesting results, mainly related to the specific characteristics of the determinants of malnutrition in every area, that deserve to be published and commented.
Authors hope that our comment have clarified the general concerns raised but If you would like us to specify any further precise points that need to be clarified, modified or added please let us know.
Reviewer 2 Report (New Reviewer)
Review Report of the research paper entitled: “Evaluation of School Children Nutritional Status in Ecuador using Nutrimetry. A Proposal of an Education Protocol to Address the Determinants of Malnutrition.” Overall Evaluation The paper is very comprehensive and provides a well-formulated proposal to address the determinants of malnutrition in Ecuador, with a hope to be evaluated and considered in other countries. It is of interest to evaluate the nutrition status of adolescents using “Nutrimetry”. Overall, I enjoyed reading this manuscript, although the authors still need to perform some amendments to this paper. Title: 1. Please correct the word “address.” Abstract: Summarized-well and contains all the information which the reader needs. Materials and Methods: 1. It is not clear what sampling method had been used to recruit children from their schools. Please be more specific. 2. I suggest removing the part of the ethical approval into a separate subheading. 3. Figure 3: please correct the word “primary”. Results: 1. I could not understand why you are justifying your findings in the result section. For instance, lines 286, 287, and 288 are debating the possible causes of observing low bottle water consumption in the Pallatanga and Penipe regions. Please delete or move these explanations to the discussion section of your manuscript. 2. I prefer to show the observed p-values in Table 2 rather than only mentioning that all pvalues were greater or less than 0.05. This is also true for Tables 3,4, and 5. Discussions 1. Lines 355 to 358. Redundant data, as you already mentioned these as study limitations. 2. Line 423. I suggest replacing the word “them” with “these parasites”. 3. Study limitations: Would not be better to say “non-representative study sample” instead of “low participation?” The paper also needs some English corrections. Finally, after managing these comments, this manuscript may be worthy of publication.

Author Response
NUTRIENTS 1873439
Evaluation of School Children Nutritional Status in Ecuador using Nutrimetry. A Proposal of an Education Protocol to Address the Determinants of Malnutrition
REVIEWER 2
ROUND 1
“The paper is very comprehensive and provides a well-formulated proposal to address the determinants of malnutrition in Ecuador, with a hope to be evaluated and considered in other countries. It is of interest to evaluate the nutrition status of adolescents using “Nutrimetry”. Overall, I enjoyed reading this manuscript, although the authors still need to perform some amendments to this paper.”
Dear Sir or Madam,
Thank you for the comprehensive review. The comments were very helpful.
We have implemented the following changes based on the revisions we received, you can see in the attached document

Round 2
Reviewer 2 Report (New Reviewer)
The manuscript can now be accepted for publications
This manuscript is a resubmission of an earlier submission. The following is a list of the peer review reports and author responses from that submission.
Round 1
Reviewer 1 Report
Dear Authors,
Thank you for this well written paper. Please note below comments and edits for your consideration.
Title: Could the title be edited to be more concise and representative of the research - mapping of childhood nutritional status.... using nutrimetry to ...
Line 115, is there are more recent Map of Poverty and Inequality by Consumption, this version is dated 2014, and may impact the discussion in the paper
Line 132, were any renumeration or incentives for parental or pupil time provided? If so, how/amount?
Line 214, could a figure illustrating the number of children recruited and participated be included, including a participation rate.
Line 379, do the results obtained truly demonstrate the importance of diagnosis prior to any type of intervention? Should nutrition policies not be implemented at all without data? Is this data representative of the general child population in Ecuador?
Line 390, please provide more information about "One Health" and why it is important. What about interventions such as mandatory reformulation, front-of-pack labelling which are most likely to have a greater impact than education - especially with the nutrition transition noted in the paper?
Line 396, all of these elements are important, is there supporting evidence from this research to inform how to take into account gender, environment, customs, beliefs and traditions?
Line 450, how are the indicators determined, are these evidence based?
In the discussion please address the following:
- how will the research inform nutrition interventions?
- how much does the cost, is it practical to complete this in all schools, regions to inform policy?
- limitations of this research, for example, the participants were selected by leaders, and other biases present. Are the participants representative of the child population?
Line 484, COI states the funder had no role in the design of he study etc... but in Line 475 it indicates there was no funding? Please clarify.
Reviewer 2 Report
· General comments
This paper as merit as it uses nutrimetry as part of the initial assessment of school nutrition and proposes an innovative framework for school nutrition intervention process. However, the paper requires strengthening in order for its scientific and programmatic contribution to be enhanced.
· Specific comments
1. Our opinion is that the general nutritional status assessment of the students is one component of information for the diagnosis of the school nutrition situation, not pre-diagnosis. Therefore, it would be of value to add the information on the other components, and even to identify the health-promoting resources of the schools in order to propose interventions, thereby testing the framework or model. Otherwise, the framework appears too theoretical. The framework, in our view, does not belong to the discussion, but to the methods.
2. The proposed intervention logical model does not show much relationship with the data collected in the study. Additionally, the last objective is actually the sought impact of the intervention. Table 7 (intervention model) could be a supplementary table as it is not essential to the paper.
3. One wonders why nothing is mentioned regarding the Nutrition-Friendly School Initiative of WHO which addresses the double burden of malnutrition and provides very detailed tools for the initial diagnosis of the school nutrition situation.
4. There should also be a short review of school nutrition interventions in the introduction.
5. The authors do not explain why they studied intestinal parasites but did not consider other nutritional status parameters that are easily assessed, such as iron or iodine status. A section on limitations of the study is essential.
6. The authors do not show clearly how the “One Health” approach is used or relevant in their study.
7. Minor comments:
a. ‘Nutricodes’ and ‘nutrimetry’: if used in the abstract, they would have to be defined.
b. The authors use too many parentheses
c. Tables 2 and 3 could be combined
d. Unless referring to both undernutrition AND overweight, avoid using ‘malnutrition’ to refer to undernutrition – see p. 2, lines 50, 75
e. How were the socio-economic variables selected?
f. Use boys and girls instead of ‘men’ and ‘women’
g. Such detail on the values of the parameters is not need, p. 5, line 172-…
Round 2
Reviewer 1 Report
Dear authors,
Thank you for taking into consideration my comments and addressing these throughout your paper.
Author Response
Thank you for your detailed review and your positive overall comments on our
manuscript.
The manuscript has undergone English language editing by MDPI. The text has been checked for correct use of grammar and common technical terms, and edited to a level suitable for reporting research in a scholarly journal (Please seethe attachment).

Reviewer 2 Report
The revised manuscript is better but there are still concerns:
1. The 'One Health' approach encompasses human, animal and environmental health concerns. Therefore, it cannot be stated that the 'One Health' approach is used.
2. The theoretical framewokr should be moved to the Methods section nd the description shortened.
3. The veryn low participation rate is a serious limitation of the study. Additionally, what is missing is an assessment of the anaemia status, which can be associated with the intestinal parasitism status.
4. In the discussion, using (or not) the NFSI tools for the initial assessment of the school nutrition situation has to be referred to.
5. Sections that can be deleted as non-relevant: Lines 486-94; Lines 516-19; 528-32.
6. A final statement could pertain to the needs for the nutrition program, perhaps again referring to NFSI, as well as the intervention planning matrix.
7. The title: It is not really 'mapping', and 'school children' should replace 'childhood'.
8. Other minor points:
a) Abstract: delete 'prediagnosis' (line 27)
b) Line 136: 'Housing' instead of 'physical';
c) Table 4: The Nutricodes have to be defined at least in a footnote;
d) Table 5 is the same as Table 4: Please correct.
Author Response
Dear Sir or Madam,
Thank you for the comprehensive review. The comments were very helpful.
We implemented the following changes based on the review we received:
We changed the TITTLE OF THE MANUSCRIPT: Evaluation of School Children Nutritional Status in Ecuador using Nutrimetry. A Proposal of an Education Protocol to Address the Determinants of Malnutrition
We MOVED THE FRAMEWORK TO THE MATERIAL AND METHODS SECTION and its description has been shortened
Detailed answers to your comments are given in the table attached.
We hope that we have been able to implement the changes satisfactorily and provide equally satisfactory responses to your comments.

Round 3
Reviewer 2 Report
The main contribution of this paper is perhaps the use of nutrimetry. However, our opinion is that there are weknesses in this study which cannot be corrected:
1. Several concepts or strategies are mentioned but they are not really addressed in the study:
a) 'One Health' (which involved human, animal, and environmental health);
b) The Nutrition Friendly School Initiative (NFSI), which was not even mentioned in the first version of the paper and the tools of which now appear to have been used; and
c) 'Taking account of gender, environment and culture': we do not see this.
2. At least all components of Phase 1 of the process should be presented in the paper, which is not the case; This is why the paper is incomplete. All that was done was anthropometry and stool analyses, and the assessment of some socioeconomic parameters.
3. As mentioned previously, the response rate was extremely low without explaining why but which hampers external validity.